# New clusterization of global seaport countries based on their DEA and FDEA network efficiency scores

**Dineswary Nadarajan**[1,2]*, **Elayaraja Aruchunan**[3,4], **Noor Fadiya Mohd Noor**[2,4]*

1 Faculty of Business, Accountancy & Law, SEGi University, Petaling Jaya, Malaysia, 2 Institute of Mathematical Sciences, Faculty of Science, Universiti Malaya, Kuala Lumpur, Malaysia, 3 Department of Decision Science, Faculty of Business & Economics, Universiti Malaya, Kuala Lumpur, Malaysia, 4 Center for Data Analytics Consultancy & Services (UM-CDACS), Faculty of Science, Universiti Malaya, Kuala Lumpur, Malaysia

* dineswarynadarajan@segi.edu.my (DN); drfadiya@um.edu.my (NFMN)

**Data Availability Statement:** Publicly available datasets were analyzed in this study. These data can be found here: [https://databank.worldbank.org/source/world-development-indicators], [https://unctadstat.unctad.org/datacentre/

## Abstract

Global seaport network efficiency can be measured using the Liner Shipping Connectivity Index (LSCI) with Gross Domestic Product. This paper utilizes k-means and hierarchical strategies by leveraging the results obtained from Data Envelopment Analysis (DEA) and Fuzzy Data Envelopment Analysis (FDEA) to cluster 133 countries based on their seaport network efficiency scores. Previous studies have explored hkmeans clustering for traffic, maritime transportation management, swarm optimization, vessel trajectory prediction, vessels behaviours, vehicular ad hoc network etc. However, there remains a notable absence of clustering research specifically addressing the efficiency of global seaport networks. This research proposed hkmeans as the best strategy for the seaport network efficiency clustering where our four newly founded clusters; low connectivity (LC), medium connectivity (MC), high connectivity (HC) and very high connectivity (VHC) are new applications in the field. Using the hkmeans algorithm, 24 countries have been clustered under LC, 47 countries under MC, 40 countries under HC and 22 countries under VHC. With and without a fuzzy dataset distribution, this demonstrates that the hkmeans clustering is consistent and practical to form grouping of general data types. The findings of this research can be useful for researchers, authorities, practitioners and investors in guiding their future analysis, decision and policy makings involving data grouping and prediction especially in the maritime economy and transportation industry.

## 1. Introduction

Maritime shipping industry is keen with machine learning development as it can help the sector with container freight customization as well as to overcome daily problems in seaport operations. Tay et al. [1] claimed that machine learning approach is easily favoured to achieve operational efficiency and productivity as it can enhance fuel efficiency in harbour vessels.

dataviewer/US.PortCalls] Additional underlying data has also been uploaded to the Figshare repository and can be accessed with the following link: https://figshare.com/s/1301f1c92d98c426e254

**Funding:** This research was conducted under the Fundamental Research Grant Scheme project FRGS/1/2022/SS02/SEGI/03/1, funded by the Ministry of Higher Education, Malaysia, headed by D. Nadarajan (SEGi University) and supervised by N.F.M. Noor (Universiti Malaya) as one of the project members. The funder had no role in study design, data collection and analysis, decision to publish, or preparation of the manuscript.

**Competing interests:** We declare that there is no competing interests in the preparation or publication of this article.

Moreover, machine learning is commonly used to estimate the travel time especially when there are congestions at the seaport.

Clustering is one of the machine learning applications that is widely used in many fields such as applied sciences, military intelligence, forensic data science, computational biology, bioinformatics, business and marketing, computer science and social science. It is a strategy that conveys information in significant clusters for the purpose of data grouping. K-means is one of the famous clustering algorithms which is broadly used since it minimizes the squared distance between two points within the same cluster [2]. K-means algorithm is superiorly applied based on the initial selection of the k-means center for more accurate and meticulous clustering. According to Dhamecha [3], k-means clustering algorithm progresses in large data-set applications through minimization of the total squared error for accuracy improvement. Just like other typical numerical methods, as the number of iteration increases, the computation time will increase as well in the k-means algorithm [4].

According to Lukauskas and Ruzgas [5], regardless of the fact that there are numerous clustering methods, the subject addressed remains as a complex matter. There is a great need for alternate procedures because typical clustering algorithms do not commonly work well with all types of datasets. Despite being one of the most common algorithms for rapid and successful implementation with certain sorts of data, there are still ample rooms for improving the accuracy of hierarchical clustering strategies. In fact, there are numerical values to indicate the level of similarity between two different hierarchical strategies when comparing them. These numerical figures are beneficial for evaluating the existing hierarchical clustering strategies [6]. On the other hand, recent developments have made vessel trajectory prediction one of the most important areas to optimize maritime transportation safety, intelligence and efficiency. It provides an up-to-date evaluation of available methodologies for vessel trajectory prediction which include the state of the art deep learning [7]. Hence, further improvement on the k-means, hierarchical and hybrid hierarchical clustering techniques are important to shape this state of the art deep learning for future smart maritime prediction.

The existing literatures revealed that majority of studies did not address hierarchical k-means (hkmeans) clustering strategy in grouping seaport network efficiency scores. As a result, the present study ventures on using the three different machine learning approaches to determine which clustering method is the most suitable for global seaport network efficiency clustering. This study introduces hkmeans where despite the algorithm itself is not new, its application in the seaport network efficiency assessment based on Liner Shipping Connecting Index output is new for 133 countries that are presently considered. This study contributes significantly to maritime research by extending the analysis and providing a comprehensive understanding of the relationship between connectivity and global economic stability which fills the literature gap in the maritime transportation industry. The study's novel contribution is also highlighted with the introduction of four new clusters defined as low connectivity (LC), medium connectivity (MC), high connectivity (HC) and very high connectivity (VHC) for the seaport network efficiency clustering while discovering the behavior of new cluster dendrograms and new cluster plots with the application of the hkmeans algorithm.

Currently the data on seaport network efficiency are uncertain due to the real-world data fluctuations in maritime industry. Secondly, the clustering strategy outcome can be affected by the fluctuated data hence the result interpretation can be misled. With existing limitations in hierarchical and k-means strategies as well as utilization of fuzzy data to treat the uncertain data, these have become the motivation of present research. Other than introducing the four new clusters to group the global 133 countries (previous studies have been done on several individual countries and up to 10 top ports in Southeast Asia using k-means but none was on hkmeans), the importance of the present study lies on the finding that application of the

hkmeans clustering strategy alone can treat the uncertain data issues, with or without fuzzy approach. Additionally, this study is important since it offers insights for better maritime data analysis, investment planning, port operations and supply chain improvements through the seaport network efficiency, while promoting sustainability and progress in economic and societal realms. The remains of the paper are arranged as follows. Section 2 provides literature review of some basic concepts in k-means and hierarchical strategies as well as for the hkmeans whereas Section 3 explains the materials and methodology of this study. Section 4 provides the findings and empirical analysis of all the clustering techniques (k-means, hierarchical and hkmeans) applied on DEA and FDEA scores of the seaport network efficiency with result comparisons between all these clustering strategies (for DEA and FDEA) are provided in Section 4.5. Finally, Section 5 concludes the overall findings of this research.

## 2. Literature review

A common statistical data analysis technique called clustering is used in many fields, including bioinformatics, machine learning, image analysis, data mining and pattern recognition. Data are divided into smaller groups before they are sorted according to a distance metric within the subgroups. There are two different kinds of clustering algorithms: partitional and hierarchical clustering strategies. K-means clustering is one of the partitional methods that assigns each data element to a unique cluster and the clusters do not overlap. Several clusters are subsets of other clusters in accordance with the hierarchical strategy. These can be agglomerative (from the bottom up) or divisive (from the top down). In light of these, hkmeans is a solution using a hybrid approach by combining the hierarchical algorithm with the partitional k-means algorithm to improve the initial non-overlapping clusters of data.

K-means is a clustering approach that is used when the data is unlabelled and it utilizes the unsupervised machine learning method [3]. Clustering of fuzzy data by virtue of the k-means algorithm can be developed in the first stage to suit a cluster with similar characteristics. On the other hand, hierarchical clustering is widely used in marine traffic, pollution level, carbon dioxide emission, collision risk, waterway limit and economy competitiveness evaluation. Hierarchical clustering can be initiated based on a density function with linking algorithms. The hierarchical algorithm contains layers of grouping which adopt the unsupervised clustering approach.

Hybrid hierarchical k-means clustering, also known as hkmeans clustering, is widely used in medical industry such as in treating Eisen's yeast microarray data, protein sequence in bioinformatics field, gene expression and in many more applications but never in maritime transportation industry for seaport network efficiency. According to Liu et al. [8] and Liu et al. [9], the involvement of hierarchical clustering with k-means algorithm in sound speed profile delivers a new method for reforming the geometric model of the sea network with different ranges. This hierarchical k-means clustering is set up to overcome the innate disadvantages such as the inability of the standard hierarchical clustering to distinguish comparable cluster patterns. In maritime transportation, the proposed cluster has been utilized to treat high-dimensional historical data for modelling the vessels' behaviour [10].

Chang et al. [11] show that few countries are influencing the efficiency of another country. Hierarchical cluster analysis is used to identify trading blocs and shipping blocs based on bilateral trade intensity and liner shipping connectivity. Therefore hierarchical clustering can be smartly performed along with the applied k-means algorithm based on each country's Liner Shipping Connectivity Index (LSCI). Initially, a particular group of countries representing their liner shipping connectivity tends to stay within their own cluster where the distance of the closest factor of interest has been checked and finally, all these countries are linked together

to decide the existence of possible similar partnerships between them. A tree diagram, also known as dendrogram, can be used to represent this long chain projection of the countries' prospective separate clusters [11].

Abdulrazzak et al. [12] illustrated the feature-reduction capabilities of the k-means clustering approach. This algorithm may be started without knowing how many clusters there really are. The study contributes parameters to the model, resulting in a more successful clustering strategy that can determine the optimal number of clusters and perform feature reduction of new hybrid clustering techniques for vehicular ad hoc network. The development of globally connected clusters will improve the high-speed railway system's transport network efficiency. The performance provided by the high-speed railway system can reduce travel time and expenses [13]. Wang et al. [14] mainly focused on cluster distribution of nodes in accordance with vectors produced after two layers of Graph Convolutional Network (GCN) was initiated. Rozar et al. [15] decided to utilize the k-means method to conduct this investigation. In order to evaluate competitiveness, a number of performance analyses were conducted using 18 bulk terminals in Malaysia that were split into two different groups with distinction in the hierarchical clustering approaches used.

The top ten container ports in Southeast Asia may be divided into three groups using k-means clustering. Nguyen and Woo [16] found that Singapore is still the region's leading port, despite competition from Port Klang, Tanjung Pelepas and Saigon Newport. A port must have stronger connections to other container ports and higher container throughput in order to be recognized as a hub port [17]. This shows that, although k-means clustering has been used in maritime transportation, 10 ports are very less as compared to present's 133 countries' ports in global hub port clustering study. The hkmeans clustering approach has been used to cluster typical scenarios of the island power supply system [18]. Only a limited number of study has been done on hkmeans clustering and that too was very far from the topic concerned presently on global seaport network efficiency clustering. Recently, a clustering algorithm with features and robust scaling for clustering ship AIS data derived using Hausdorff distance and Hierarchical Density-Based Spatial Clustering of Applications with Noise (HDBSCAN), was suggested by Wang et al. [19]. According to Andrade et al. [20], the top five most efficient ports are those with the highest cargo throughput and it shows a significant link between cargo throughput and port efficiency rating among Brazilian ports. The clustering algorithm classified the Brazilian ports into three categories: efficient, moderately efficient and inefficient. This again shows that the study was conducted only for a single country's ports and the outcome gives three efficiency clusters.

Martinez-Budrai et al. [21] used DEA scores of 26 Spanish port authorities to divide the ports' levels of complexity into three categories. Following this, Quresma Dias et al. [22] focussed on 10 Iberian Peninsula container terminals while Guironnet et al. [23] examined technical efficiency of 24 Italian and 13 French ports using DEA and clustered the ports into geographical grouping. Similarly, Sharma and Yu [24], Koster et al. [25], Cheon [26], Cullinane and Wang [27], Wu and Goh [28], Cheon et al. [29] and Bichou [30] used DEA to assess technical efficiency of 70, 38, 110, 25, 21, 98 and 60 global container terminals, respectively. Afterwards, terminal clusters obtained from Serviceable Obtainable Market (SOM) and local competition were grouped based on ownership and corporate change by Cheon et al. [29]. Tovar and Rodrguez-Déniz [31] clustered 26 Spanish port authorities using the dendrogram cutoff in hierarchical clustering. The present literature survey reveals that all the past studies predicted technical efficiency using the DEA model and only two studies utilized hierarchical clustering.

The results, based on Zhanjiang Port [9], show that the hybrid clustering technique can effectively cluster ship trajectories and provides categorization of ship traffic. Yet, the

effectiveness of the seaport network based on LSCI and GDP output has never been studied using hkmeans clustering. The majority of researches done had focused on traffic, maritime transportation management, swarm optimization, vessel trajectory prediction, vessels behaviours, vehicular ad hoc network etc., but there has not been a single clustering work on the effectiveness of the seaport network by comparing various strategies (k-means, hierarchical and hkmeans) using four presently defined clusters (LC, MC, HC, VHC). The absence of accuracy has initiated a combination of DEA model with fuzzy set theory where it results in FDEA data to Tackles the existence of outputs deemed undesirable [32]. In order to leverage these efforts, the present paper proposed the hybrid hkmeans strategy in clustering the seaport network efficiency scores of 133 countries obtained from DEA and FDEA where comparisons have been done between the results of k-means, hierarchical and hkmeans techniques. Since hkmeans clustering on seaport network efficiency based on LSCI and GDP output was never done in the past, it creates motivation for the present study. Moreover, the introduction of the four new level clusters with different specifications through this research is important for the global maritime industry as the findings on seaport network efficiency contribute towards the country's efficiency, hence the country's economic growth.

The present research's main contributions are clarified as follows:

1. This study introduces four new level clusters specified as low connectivity (LC), medium connectivity (MC), high connectivity (HC) and very high connectivity (VHC) in clustering the seaport network efficiency scores. The nearest study by Andrade et al. [20] categorized leading Brazilian ports based on cargo throughput efficiency, dividing them into only three groups (highly efficient, moderately efficient and inefficient).

2. This study utilizes DEA and FDEA in the countries' seaport network efficiency measurement prior to the conduct of clustering. None of the previous studies including [21–31] have considered both DEA and FDEA in the clustering strategy applications at all.

3. This study clusters 133 global seaport countries which is the highest number of countries considered in similar research area. Previously Nguyen and Woo [16], employed k-means cluster analysis for top 10 Southeast Asian ports' countries.

4. This study applies k-means and hierarchical strategies as well as recommends the third strategy, hkmeans (hierarchical k-means) for seaport network efficiency clustering. In the nearest past study, k-means strategy has been applied in social network analysis within the maritime transportation context, focusing on the top 10 Southeast Asian ports' [16].

5. The present study implements LSCI and GDP output in both DEA and FDEA computations prior to the clustering application which was never been carried out before. Previously, Chang et al., [11] has only used LSCI to perform hierarchical clustering strategy.

## 3. Materials and methodology

### 3.1 Data sources and variables

The seaport network efficiency scores are calculated based on four input variables (time in port, age of vessels, size of vessels and cargo carrying capacity) and two output variables (gross domestic product (GDP) and liner shipping connectivity index (LSCI)) of 133 global seaport countries listed in Table 1. The input variables are collected from United Nations Conference on Trade and Development statistics (UNCTADstat) and the output variables are from World Development Indicators (WDI). The links to the data source of the UNCTADstat and WDI

**Table 1. 133 countries presently considered for seaport network efficiency clustering.**

| No | Country | No | Country | No | Country | No | Country |
|---|---|---|---|---|---|---|---|
| 1 | Albania | 35 | Djibouti | 69 | Korea (Republic of) | 103 | Portugal |
| 2 | Algeria | 36 | Dominica | 70 | Kuwait | 104 | Qatar |
| 3 | American Samoa | 37 | Dominican Republic | 71 | Latvia | 105 | Romania |
| 4 | Angola | 38 | Ecuador | 72 | Lebanon | 106 | Russian Federation |
| 5 | Antigua and Barbuda | 39 | Egypt | 73 | Liberia | 107 | Samoa |
| 6 | Argentina | 40 | El Salvador | 74 | Libya | 108 | Saudi Arabia |
| 7 | Australia | 41 | Estonia | 75 | Lithuania | 109 | Senegal |
| 8 | Bahamas | 42 | Fiji | 76 | Madagascar | 110 | Seychelles |
| 9 | Bahrain | 43 | Finland | 77 | Malaysia | 111 | Sierra Leone |
| 10 | Bangladesh | 44 | Gabon | 78 | Maldives | 112 | Singapore |
| 11 | Barbados | 45 | Gambia | 79 | Malta | 113 | Solomon Islands |
| 12 | Belgium | 46 | Georgia | 80 | Mauritania | 114 | Somalia |
| 13 | Belize | 47 | Germany | 81 | Mauritius | 115 | Spain |
| 14 | Benin | 48 | Greece | 82 | Mexico | 116 | Sri Lanka |
| 15 | Brazil | 49 | Grenada | 83 | Micronesia (Federated States of) | 117 | Sudan |
| 16 | Brunei Darussalam | 50 | Guam | 84 | Moldova (Republic of) | 118 | Suriname |
| 17 | Bulgaria | 51 | Guatemala | 85 | Montenegro | 119 | Sweden |
| 18 | Cambodia | 52 | Guinea | 86 | Morocco | 120 | Tanzania |
| 19 | Cameroon | 53 | Guinea-Bissau | 87 | Mozambique | 121 | Thailand |
| 20 | Canada | 54 | Guyana | 88 | Myanmar | 122 | Timor-Leste |
| 21 | Cayman Islands | 55 | Haiti | 89 | Namibia | 123 | Togo |
| 22 | Chile | 56 | Honduras | 90 | Netherlands | 124 | Tonga |
| 23 | China | 57 | Iceland | 91 | New Zealand | 125 | Trinidad and Tobago |
| 24 | China, Hong Kong SAR | 58 | India | 92 | Nicaragua | 126 | Tunisia |
| 25 | Colombia | 59 | Indonesia | 93 | Nigeria | 127 | Turkey |
| 26 | Comoros | 60 | Iran (Islamic Republic of) | 94 | Norway | 128 | Ukraine |
| 27 | Congo | 61 | Iraq | 95 | Oman | 129 | United Arab Emirates |
| 28 | Congo (Dem. Rep. of) | 62 | Ireland | 96 | Pakistan | 130 | United Kingdom |
| 29 | Costa Rica | 63 | Israel | 97 | Panama | 131 | United States of America |
| 30 | Côte d'Ivoire | 64 | Italy | 98 | Papua New Guinea | 132 | Uruguay |
| 31 | Croatia | 65 | Jamaica | 99 | Paraguay | 133 | Viet Nam |
| 32 | Cuba | 66 | Japan | 100 | Peru | | |
| 33 | Cyprus | 67 | Jordan | 101 | Philippines | | |
| 34 | Denmark | 68 | Kenya | 102 | Poland | | |

are provided under this work's data availability statement. In this study, data from 133 countries with seaports was collected, comprising input and output variables to assess the efficiency of the seaport networks. This assessment was conducted using both Data Envelopment Analysis (DEA) and Fuzzy Data Envelopment Analysis (FDEA), incorporating triangular fuzzy number theory. The study utilized MaxDEA software to compute seaport network efficiency scores. In a prior study [33], it was established that when dealing with both real and fluctuating

data, the utilization of triangular fuzzy numbers is proven to be more proficient than trapezoidal fuzzy numbers for calculating efficiency scores for FDEA.

Firstly, a linear programming (LP) problem is formulated as follows [33]:

$$max \ F = \frac{\sum_{g=1}^{h} u_g y_{gj}}{\sum_{k=1}^{n} v_k x_{kj}}, \tag{1}$$

subject to:

$$\frac{\sum_{g=1}^{h} u_g y_{gj}}{\sum_{k=1}^{n} v_k x_{kj}} \leq 1, \quad j = 1, \ldots, z, \tag{2}$$

$$\frac{u_{gj}}{\sum u_g y_{gj}} \geq \varepsilon, \frac{v_{kj}}{\sum v_k x_{kj}} \geq \varepsilon \quad g = 1, \ldots, h \text{ and } k = 1, \ldots, n. \tag{3}$$

To make this LP viable for DEA, Eq (1) until Eq (3) are reformulated as follows:

$$max \sum_{g=1}^{h} u_g y_{gj} \tag{4}$$

subject to:

$$\sum_{k=1}^{n} v_k x_{kj} = 1, \tag{5}$$

$$\sum_{g=1}^{h} u_g y_{gj} - \sum_{k=1}^{n} v_k x_{kj} \leq 0, \tag{6}$$

$$u_g, v_k \geq 0, \ g = 1, \ldots h \text{ and } k = 1, \ldots, n. \tag{7}$$

On the other hand, Eq (1) until Eq (3) can also be modified to allow rooms for fuzzy numbers with inclusions of $L$ (minimum value), $A$ (mean value) and $M$ (maximum value) to form the following LP:

$$max \frac{\sum_{g=1}^{h} \left( u_g^L y_{gj}^L + u_g^A y_{gj}^A + u_g^M y_{gj}^M \right)}{\sum_{k=1}^{n} \left( v_k^L x_{kj}^L + v_k^A y_{kj}^A + v_k^M y_{kj}^M \right)} \tag{8}$$

subject to

$$\frac{\sum_{g=1}^{h} \left( u_g^L y_{gj}^L + u_g^A y_{gj}^A + u_g^M y_{gj}^M \right)}{\sum_{k=1}^{n} \left( v_k^L x_{kj}^L + v_k^A y_{kj}^A + v_k^M y_{kj}^M \right)} \leq 0, \tag{9}$$

$$v_k^L, u_g^L \geq 0, v_k^A, u_g^A \leq 0, v_k^M, u_g^M \quad 1 \leq k \leq n, 1 \leq g \leq h. \tag{10}$$

Similarly, Eq (8) until Eq (10) can be reformulated to fit FDEA as follows:

$$max \sum_{g=1}^{h} \left( u_g^L y_{gj}^L + u_g^A y_{gj}^A + u_g^M y_{gj}^M \right) \tag{11}$$

subject to

$$\sum_{k=1}^{n} \left( v_k^L x_{kj}^L + v_k^A y_{kj}^A + v_k^M y_{kj}^M \right) = 1, \tag{12}$$

$$\sum_{g=1}^{h} \left( u_g^L y_{gj}^L + u_g^A y_{gj}^A + u_g^M y_{gj}^M \right) - \sum_{k=1}^{n} \left( v_k^L x_{kj}^L + v_k^A y_{kj}^A + v_k^M y_{kj}^M \right) \le 0, \tag{13}$$

$$v_k^L - v_k^A \le -\varepsilon, v_k^M - v_k^A \le -\varepsilon, v_k^L v_k^A v_k^M \ge 0, 1 \le k \le n, \tag{14}$$

$$u_g^L - u_g^A \le -\varepsilon, u_g^M - u_g^A \le -\varepsilon, u_g^L u_g^A u_g^M \ge 0, 1 \le g \le h. \tag{15}$$

The results of DEA (from Eq (4) until Eq (7)) and FDEA (from Eq (11) until Eq (15)) based on 3-year available public data (2018–2020) are then used to perform the clustering. Three clustering strategies are explored in this current work; k-means, hierarchical and hierarchical k-means (or hkmeans). These algorithms are coded in RStudio software using R-programming to construct four new clusters for grouping the 133 countries based on their seaport network efficiency data of DEA and FDEA obtained previously. Further elaboration and comparison between the three different clustering strategies are presented in the next sections following the stepwise manner.

Fuzzy Data Envelopment Analysis (FDEA) is a method used to assess the efficiency of decision-making units (DMUs) when dealing with uncertain or imprecise data [34]. Its advantages include:

1. Handling uncertainty: FDEA accommodates uncertain data by allowing for degrees of membership, making it suitable for situations with incomplete or noisy data

2. Flexibility: It provides flexibility in modeling input-output dynamics, encompassing diverse performance-contributing factors, particularly in scenarios where quantification presents challenges.

3. Robustness: FDEA is robust against outliers and extreme values, producing more reliable efficiency scores, especially in complex systems with variable data.

4. Accounting for subjectivity: It captures subjective judgments and expert opinions, incorporating qualitative factors into evaluations.

5. Interpretability: FDEA provides easily interpretable results, identifying relative efficiency of DMUs and areas for improvement.

FDEA was chosen for seaport network efficiency in this study because it can handle uncertain data, common in maritime contexts. It incorporates fuzzy set theory, allowing for alternate evaluations of efficiency amidst the complexities and uncertainties of the data.

### 3.2 Methodology

**3.2.1 K-means clustering.** In this section, the step-by-step procedures to perform k-means clustering are briefed. There are four steps to conduct the k-means algorithm [35].

*Step 1*: Determination of the k-value: A number of clusters to be used in the study is selected randomly as the underlying initial cluster communities.

*Step 2*: Finding the nearest centroid: The nearest centroid is based on the Euclidean distance between the observation and the centroid. The Euclidean distance between two points $a(x_1, y_1)$ and $b(x_2, y_2)$ is given as in Eq (16):

$$d(a, b) = \sqrt{(x_1 - x_2)^2 + (y_2 - y_2)^2}. \tag{16}$$

*Step 3*: For each k-means cluster, a new mean value of all data considered is recalculated using Eq (17) where Pi is the set of all observations allocated to the *i*-th cluster:

$$c_i = \frac{1}{|P_i|} \sum x_i. \tag{17}$$

*Step 4*: Steps 2 and 3 are repeated until the total sum of squares is minimized and the centroids are no longer changed or the maximum iteration has been reached.

**3.2.2 Elbow method.** An essential component of this approach is to determine the appropriate number of clusters. Elbow method is a widely used technique for determining the appropriate k-value [35]. The elbow approach is a heuristic method commonly used in cluster analysis to estimate the number of clusters present in a dataset. Plotting the explained variation as a function of the number of clusters, the procedure entails towards choosing the elbow of the curve as the appropriate number of clusters.

**3.2.3 Hierarchical clustering.** The hierarchical clustering is performed as the second objective of this study. This strategy measures the distance to generate new clusters. The procedures are branched into 5 steps [35].

*Step 1*: The distances between each pair of points using a distance metric is determined.

*Step 2*: Each data point is assigned to a cluster.

*Step 3*: The grouping is constructed based on close similarity between one another.

*Step 4*: The distance metric is refreshed.

*Step 5*: Step 3 and 4 are repeated until a single cluster is obtained.

**3.2.4 Hierarchical k-means clustering.** The hkmeans strategy is carried as follows [35]:

*Step 1*: Hierarchical clustering is performed.

*Step 2*: K-clusters are divided by cutting the tree.

*Step 3*: The closest centroid is determined by averaging each cluster.

*Step 4*: K-means algorithm is performed by using the initial cluster centers from the set of centroids calculated in Step.

Fig 1 illustrates the step-by-step data collection process leading to clustering. Initially, four input variables were gathered from UNCTADstats, along with two outputs from WDI. Subsequently, the data underwent a screening to eliminate outliers for normalizing the data. Once the screened data was ready, Data Envelopment Analysis (DEA) will be executed using Max-DEA. Afterward, the screened data was utilized for Fuzzy Data Envelopment Analysis

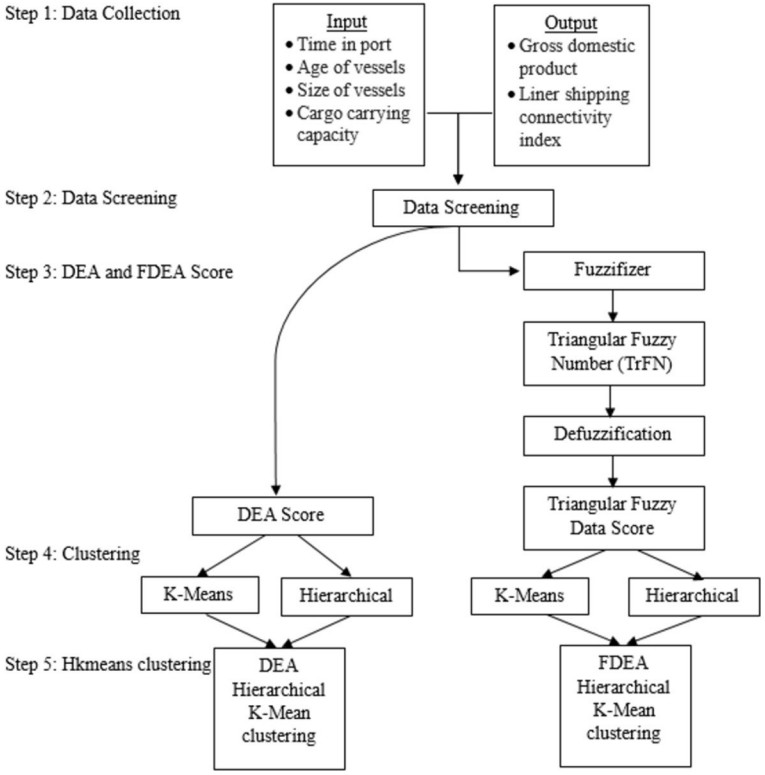

**Fig 1. Methodologies.**

(FDEA), involving data fuzzification to generate Triangular Fuzzy Number (TFN), followed by defuzzification to derive the ultimate FDEA scores. To cluster countries by efficiency, both DEA and FDEA scores were subjected to clustering through k-means and hierarchical strategies using R-programming. Additionally, for addressing fluctuations in maritime data, the hkmeans clustering strategy was applied. The research will be concluded by comparing the outcomes obtained from all the three clustering methods considered.

## 4. Results and empirical analysis

### 4.1 Outlier detection

The work starts with outlier identification since clustering is very sensitive to outliers and clustering can only be done if the data is free from any outliers. Therefore, the specific box plot is drawn to check the outlier as shown in Fig 2. There are no outliers identified in the results of DEA and FDEA from the boxplot, hence it proves the inexistence of any extreme value.

### 4.2 K-means clustering results

K-means clustering is performed by leveraging the seaport network efficiency scores obtained from DEA and FDEA. Further analysis and comparison between the two datasets can be performed after the k-value is determined prior to finding the nearest centroid. K-means clustering is developed in this study where it calculates the sum of square and the average of distance of points in the seaport network efficiency.

The present study utilizes the elbow method which can guide the way to find the best k-cluster value of the data. A plot is developed with a number of clusters and sum of square

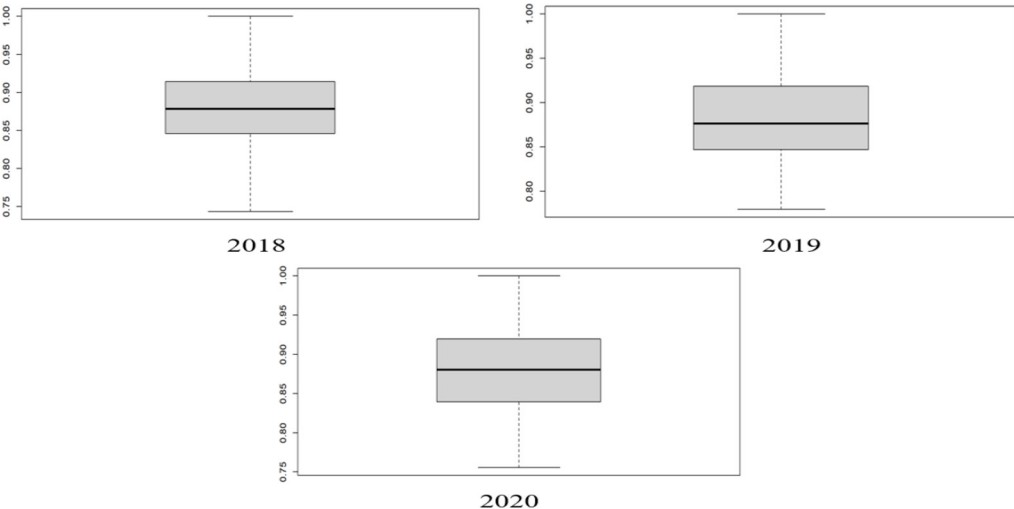

**Fig 2. Boxplot of outlier detection.**

within the cluster to obtain the k-value. All the three-year results via the elbow method have shown that only four clusters are required to categorize the seaport network efficiency scores based on Fig 3. The four clusters can be determined by observing the graph in details and the k-value can be selected at the point where the graph decreases sharply which happens at k-value of 4.

The gap statistics is the best tool to find a suitable k-value in the clustering. In-vented 20 years back by Tibshirani et al. [36], the graph is interpreted based on the break point within the cluster distance to obtain the optimum cluster. Figs 3 and 4 visualize the number of clusters and the statistic gaps for DEA and FDEA for the two considered years (2018 and 2019). Based on these figures, it is shown that the break point happens at the clusters 3 and 4. This decision will be verified by the 2020 dataset.

Fig 5 depicts the gap statistics used to determine the number of clusters. Different natures in the 2018 and 2019 datasets are depicted in this figure to find the number of clusters. The gap statistics for 2020 reveal that the number of clusters for DEA can be 3, 4, 5 or 6 but the gap statistics for FDEA are 3 or 4. This is an undesirable output with the use of DEA as compared to FDEA that might be routed from misleading interpretations caused by uncertainty and fluc-tuation in real-world data, particularly during COVID-19 pandemic. Therefore, it is evident from here that the k-means strategy is more sensitive for the FDEA dataset than the DEA data-set due to the fuzziness contribution. Hence, it can be emphasized that the k-means strategy provides better clustering for fuzzy data distribution. On the other hand, data from 2020 are irregular due to the impact of the COVID-19 outbreak on the seaport network efficiency scores.

Based on the variations in 2018, 2019 and 2020, all the three-year plots show that the num-ber of clusters of 3 and 4 are optimum. Following this result, this research will consider 4 clus-ters in each clustering approach on DEA and FDEA datasets. These 4 new level clusters are now specified as low connectivity (LC), medium connectivity (MC), high connectivity (HC) and very high connectivity (VHC).

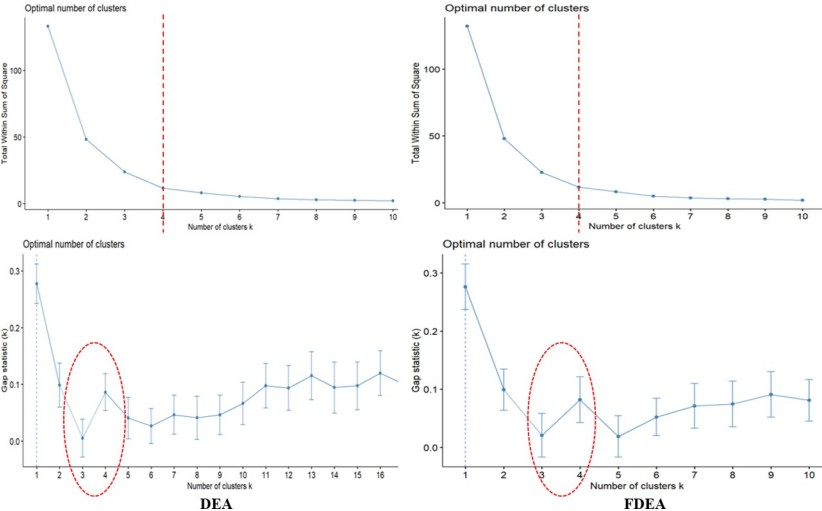

**Fig 3. Number of clusters and gap statistics for k-means clustering for 2018.**

## 4.3 Hierarchical clustering results for 2018–2020

Hierarchical clustering is another method that can cluster a set of data into groups. It is repetitively performed using two steps. The first step is to identify the two clusters that are closest together and it continues with combining the two most alike clusters. The graphs comparing DEA and FDEA for 2018, 2019 and 2020 are shown in Figs 6–8 respectively. Here the hierarchical clustering package in RStudio has automatically set 'black' as the default colour to denote the dendrogram branches and the country numbers. The hierarchical clustering starts with each country number (point) assigned to a separate cluster. The cluster is performed by combining the nearest clusters into a bigger cluster until it gives the four nearest clusters that can be displayed using the dendrogram. The cluster dendrogram shows the data points in the x-axis and in the y-axis represent the distance between the clusters. The line with green colour represents the domain for each cluster. Hierarchical cluster is a decision tree that divides the

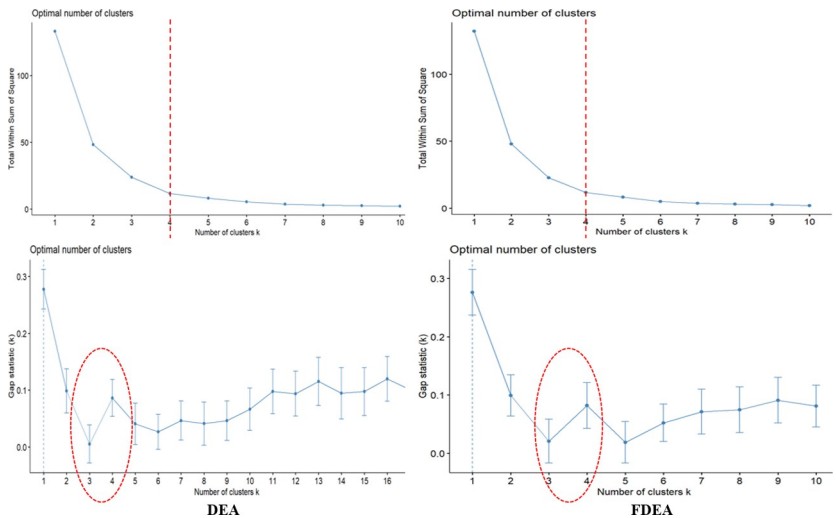

**Fig 4. Number of clusters and gap statistics for k-means clustering for 2019.**

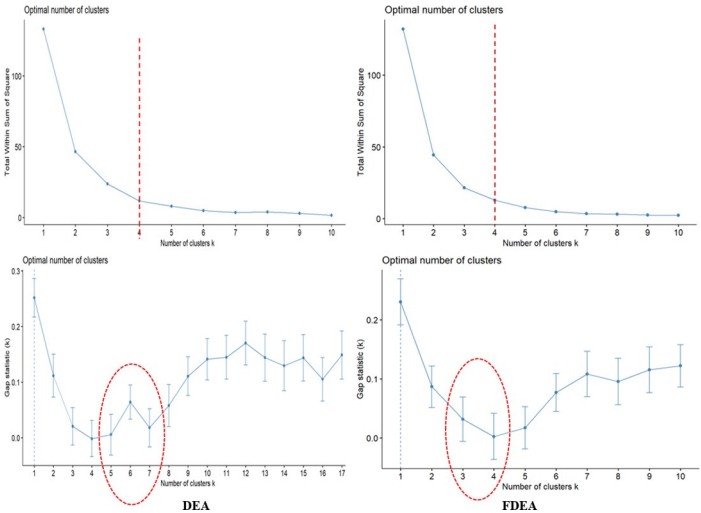

**Fig 5. Number of clusters and gap statistics for k-means clustering for 2020.**

cluster visually with intention to have a minimum distance in the y-axis. Similarly, in every splitting route of the dendrogram, the data belong to the clusters with different levels of efficiency among them. Figs 6–8 show dendrograms for DEA and FDEA datasets using the

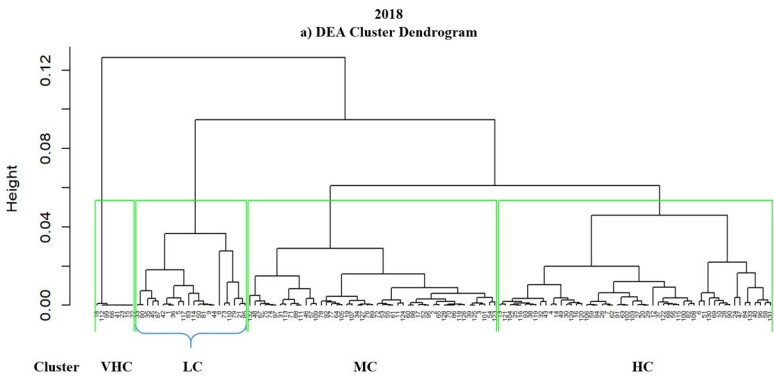

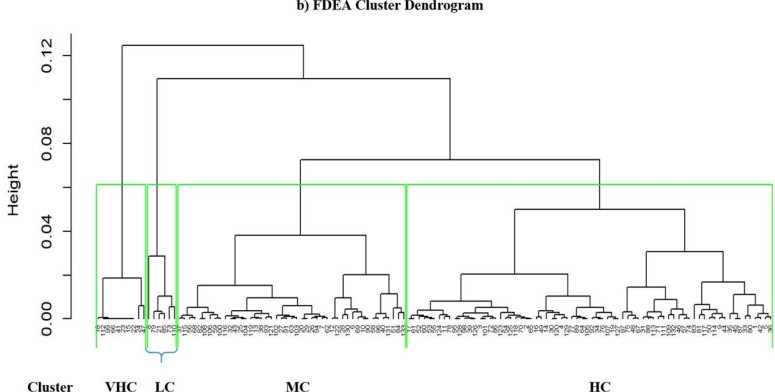

**Fig 6. Hierarchical dendrogram for DEA and FDEA for 2018.**

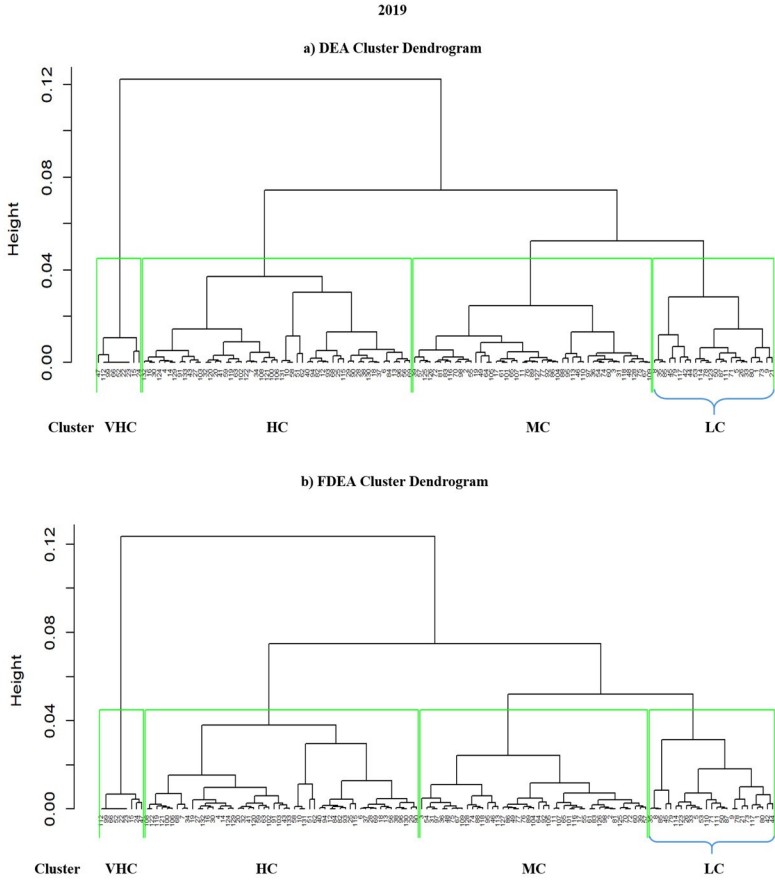

**Fig 7. Hierarchical dendrogram for DEA and FDEA for 2019.**

hierarchical algorithm. All the three-year results show that the groupings are very narrow in VHC Cluster for both DEA and FDEA. The LC cluster for FDEA dataset in 2018 and 2020 are also narrow as shown in Figs 6 and 8 respectively. Based on Fig 7, the hierarchical clustering results for DEA and FDEA show similar outputs for 2019. The diagram highlights the undesirable outcomes for Figs 6 (2018) and 8 (2020). For example, Fig 6 illustrates that under LC, there were 22 countries in 2018, whereas after employing FDEA, the number decreased to 6. Similarly, Fig 8 demonstrates that the LC cluster comprised 52 countries according to DEA, but only 3 under FDEA. These huge differences in the findings between DEA and FDEA might be subject to potential misinterpretation of data when exclusively relying on DEA, as it clustered more countries under LC for both years. These discrepancies can be attributed to uncertainty and fluctuation of the data under the global crises experienced during these years, including the intensifying trade war between the United States of America and China in 2018, and the impact of the COVID-19 pandemic in 2020.

### 4.4 Hierarchical k-means (Hkmeans) clustering results

This study further explores hkmeans strategy to optimize the clustering outputs for the seaport network efficiency scores. The novelty of this work is because there is no study in maritime industry that uses hkmeans strategy in the clustering of 133 global countries' seaport network efficiency scores. This hkmeans clustering strategy is proposed due to the drawbacks in conventional k-means and hierarchical algorithms that produce variation of results in the

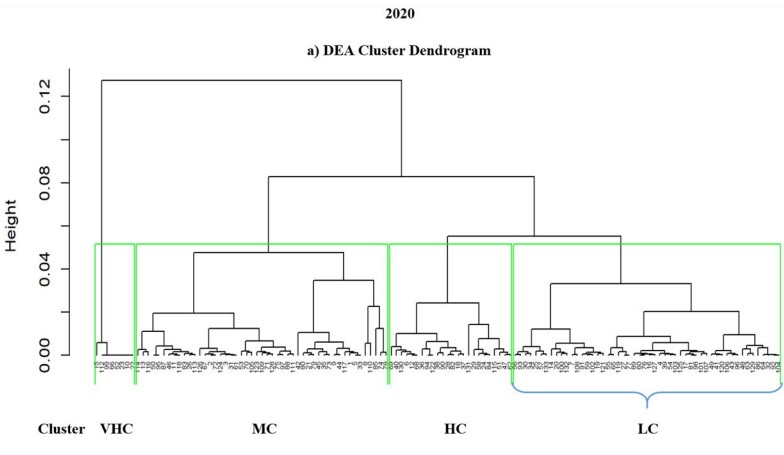

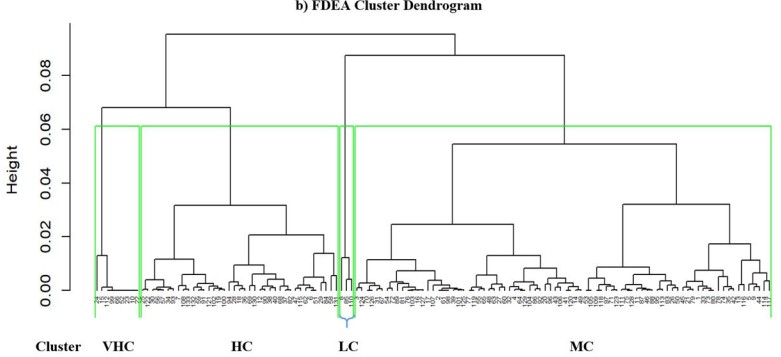

**Fig 8. Hierarchical dendrogram for DEA and FDEA for 2020.**

calculation. K-means is very sensitive to initial selection of the clusters where a random set of countries has been selected as the initial center. On the other hand, the outcomes of the hierarchical clustering strategy might be diverse with different type of dataset applied. By combining the elements of hierarchical technique with k-means technique in the hierarchical k-means or hkmeans algorithm, the advantages of these two techniques can be leveraged while their individual drawbacks can be retrenched to be more balanced. This potential motivates exploration and application of the hkmeans strategy in the seaport network efficiency clustering which can provide better consistency and practicality for general data type.

Unlike Figs 6–8 where hierarchical clustering was imposed with 'black' as the default colour for both the branches and the country numbers, there are four different colours indicating the four clusters of LC (green branch), MC (black branch), HC (red branch) and VHC (blue branch) in Fig 9. Initially, all the country numbers have similar colour with the branches as in Figs 6–8. However after imposing hkmeans clustering in Fig 9 for DEA and FDEA, some of the country numbers have been shifted under different colored branch. For instance, the country numbers 110, 8, 85, 35, 45, 79, 21, 73, 44, 117, 42, 1, 80, 87, 5, 33, 9 and 114 are originally written in blue colour under the blue branch (VHC). However, after implementing hkmenas clustering, all these country numbers are now written in green colour (LC) indicating that their cluster has been shifted from VHC to LC. Here are some more specific details from Fig 9; Brunei Darussalam is classified under MC cluster (DEA) and HC cluster (FDEA), Conga is under VHC cluster (DEA) and MC cluster (FDEA), whereas Latvia, Sierra Leone and Solomon Islands are classified from HC cluster (DEA) to LC cluster (FDEA), following the hkmeans

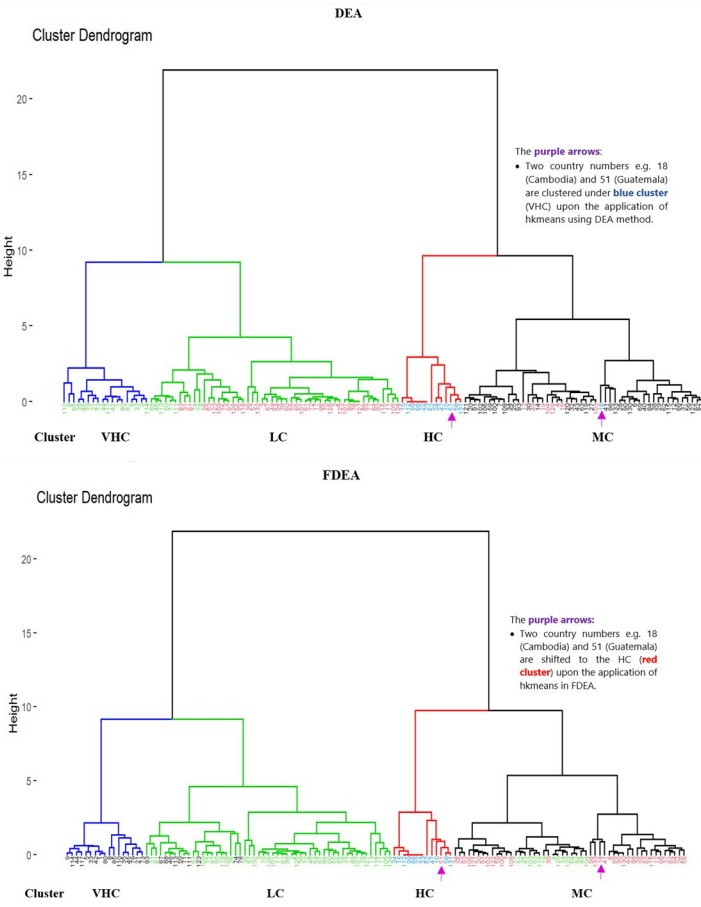

**Fig 9. Comparison of hkmeans clustering for DEA and FDEA sore.**

clustering algorithm. Moreover, Turkey is classified under HC and VHC cluster for DEA and FDEA datasets respectively. Note that only the clustering results of these 6 countries changed with the hkmeans algorithm. The clustering results for other countries remain under the same clusters with improved accuracy in cluster prediction through the integration of the hkmeans clustering technique. With and without a fuzzy dataset distribution, this demonstrates that the hkmeans clustering is consistent and practical to form grouping of general data types. Hence the hkmeans strategy is an appropriate tool for the seaport network efficiency clustering.

## 4.5 Differences in hierarchical, k-means and hkmeans clustering results

**4.5.1 Hierarchical and k-means results for 2018–2020.** The comparisons between hierarchical and k-means strategies for DEA and FDEA results are shown in Fig 10. The graph shows the difference according to the clusters; LC, MC, HC and VHC based on the seaport network efficiency obtained in the previous analysis. From the figure, the MC has the highest frequency among all the clusters. The y-axis is representing the frequency of the countries involved in this study and x-axis represents the cluster categories that are used in this study. It is observed that the results of k-means for DEA and FDEA data are approximately similar as opposite to hierarchical results for the two datasets. Moreover we found that the hierarchical clustering results also show a bit of fluctuation in 2018 and 2020 for the seaport network efficiency which indicates that the hierarchical clustering strategy is not stable as compared to the

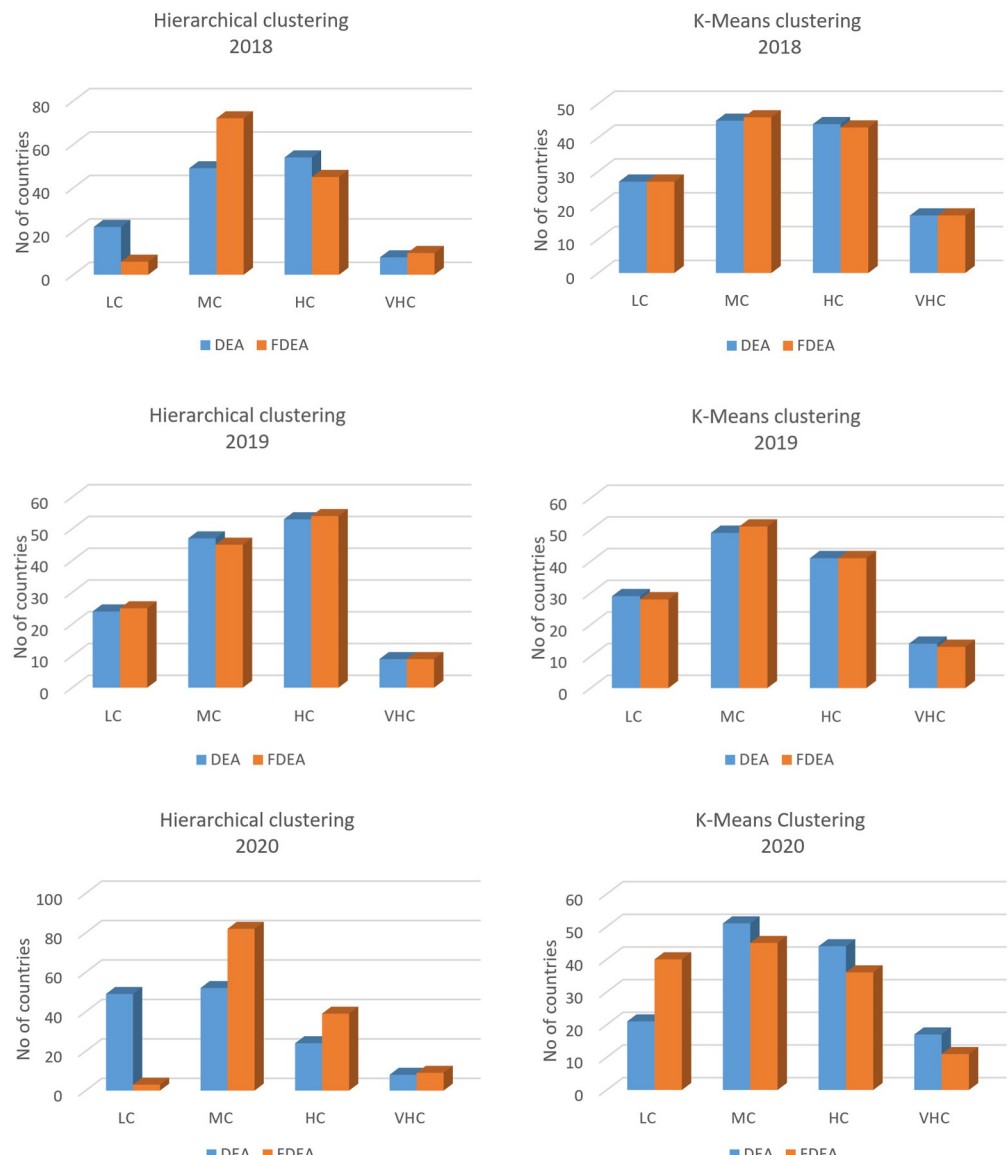

**Fig 10. Comparison between hierarchical and k-means clustering results on DEA and FDEA datasets.**

results of 2019. Therefore, between the hierarchical and k-means clustering algorithms alone from Fig 10 especially for the year of 2020, it is concluded that the k-means technique is a better strategy than the hierarchical strategy in clustering the seaport network efficiency scores since the fluctuation risk between the regular and fuzzy data distributions is minimal.

**4.5.2 Hierarchical and hkmeans results for DEA and FDEA.** Oyewole et al. [37], examined clustering algorithms, focusing on the conventional strategies of partitioning and hierarchical clustering. K-means clustering divides a dataset into k number of clusters based on proximity to centroids, necessitating a predefined k number and being sensitive to initial centroids. Hierarchical clustering constructs a hierarchical tree of clusters, uncovering nested structures without the need for a predefined k, albeit with potential computational intensity. Hkmeans integrates the flexibility of hierarchical clustering with the efficiency of k-means by utilizing hierarchical clustering for initial cluster hierarchy establishment and subsequently

refining it with the k-means strategy. Examining the differences and connections among the clusters created by each algorithm allows a deeper understanding of the fundamental patterns within the dataset. These methodologies equip insights needed to make informed decisions about choosing the most suitable clustering approach for a specific application [38].

Hkmeans method is firstly conducted by employing the hierarchical method to determine the k-value where the tree is cut into clusters. There are four seaport network efficiency clusters; LC, MC, HC and VHC represented by four coloured main tree branches as depicted in Figs 11 and 12 respectively. Under these clusters, the numbers representing the seaport countries are classified based on their seaport network efficiency level. The dendrogram of hierarchical algorithm is marked with purple, blue, green and red colours, whereas the hkmeans dendrogram is displayed in black, green, red and blue colours to represent MC, LC, HC and VHC clusters respectively in both Figs 11 and 12.

Based on Fig 11, a few countries under the hierarchical diagram have been moved from MC cluster to HC cluster (red) after utilization of the hkmeans method where these countries are Belize, Grenada, Cameroon, Philippines, United Arab Emirates, Angola, Brunei Darussalam and Saudi Arabia. Cambodia is the only country that has been moved from MC cluster of the hierarchical method to VHC cluster (blue) of the hkmeans method. From the hierarchical LC cluster (green), the majority of the countries are reassigned to hkmeans HC cluster, while the remaining countries that stay under the LC cluster through hierarchical and hkmeans

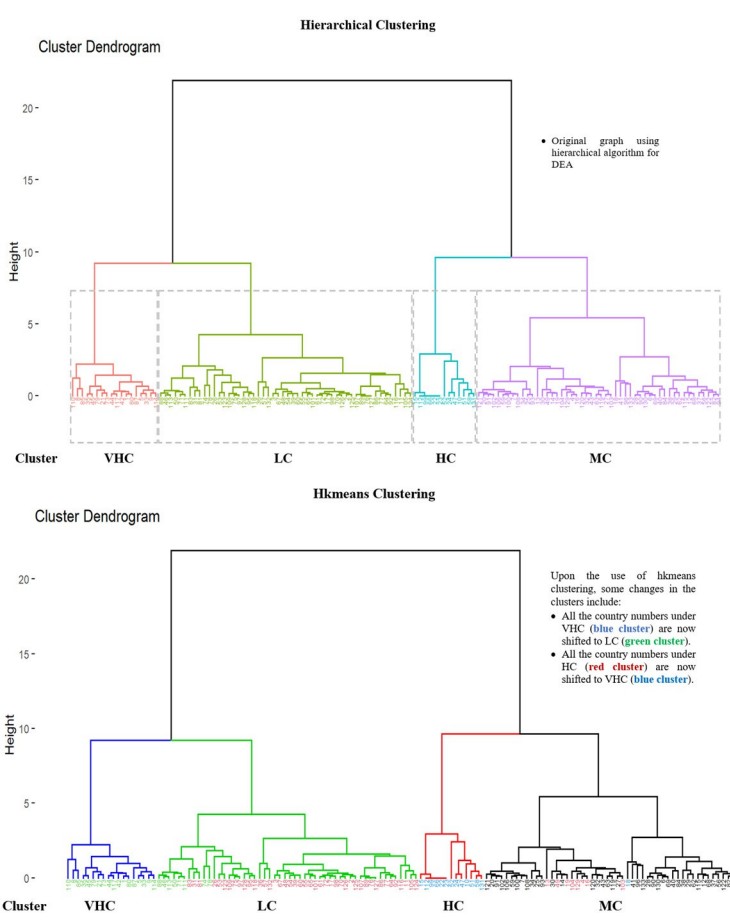

**Fig 11. Comparison of hierarchical and hkmeans results for DEA data.**

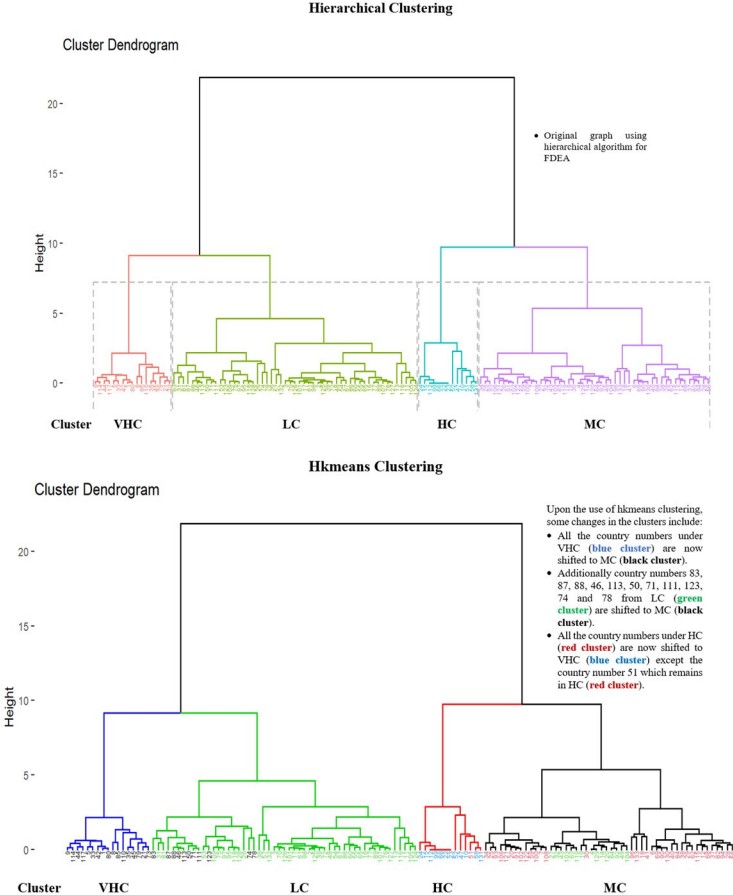

**Fig 12. Comparison of hierarchical and hkmeans results for FDEA data.**

strategies are Myanmar, Georgia, Solomon Islands, Guam, Latvia, Sierra Leone, Libya and Maldives. Last but not least, it is noticed that the hierarchical HC cluster has an intriguing feature such that all the countries under this cluster have been changed to VHC cluster of hkmeans, whereas all other countries from VHC cluster of the hierarchical clustering have been shifted to the LC cluster of the hkmeans clustering. In general, this figure shows how the hierarchical clustering results can be different from the results of hkmeans clustering.

Fig 12 shows the comparison between hierarchical and hkmeans clustering results for FDEA dataset, which illustrate that all countries under the hierarchical VHC cluster have been changed to the hkmeans MC cluster while all countries under the hierarchical HC cluster have been changed to the hkmeans VHC cluster except for Guatemala. Besides that, Micronesia, Mozambique, Myanmar, Georgia, Solomon Islands, Guam, Sierra Leone, Togo, Libya and Maldives are transferred to the hkmeans MC cluster from the hierarchical LC cluster while other countries under the hierarchical LC cluster remain in the same cluster even after the utilization of hkmeans strategy. Regarding the countries under the hierarchical MC cluster, all of them have been changed to either LC or HC cluster under the hkmeans strategy.

**4.5.3 K-means versus hkmeans for DEA and FDEA.** Figs 13 and 14 demonstrate the different clusters with k-means and hkmeans clustering strategies for both DEA and FDEA seaport network efficiency datasets. The cluster plots show that a few countries have been moved to other clusters following the use of hkmeans clustering strategy with respect to the countries'

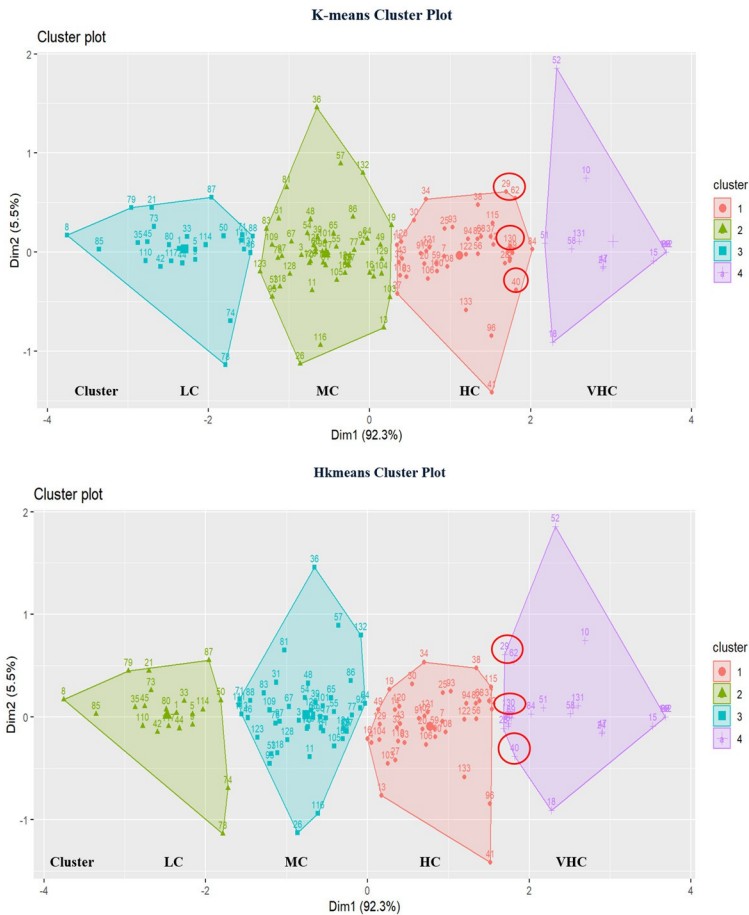

**Fig 13. Comparison of k-means and hkmeans clustering plots for DEA.**

particular efficiency characteristics. Few values are highlighted as data samples in Figs 13 and 14 to differentiate the cluster plots between the classical k-means and hkmeans clustering strategies. Based on selected data samples from Fig 13 involving the countries numbered with 29 (Costa Rica), 62 (Ireland), 130 (United Kingdom) and 40 (El Salvador), it is clearly shown that these countries have been changed and plotted into another cluster after the involvement of hkmeans algorithm. It also shows that the hkmeans clustering method produces nearly similar outcomes for DEA and FDEA datasets.

The cluster plot, which displays clusters in two-dimensional space, is shown in Fig 13 (Dim1 and Dim2). This measurement is basically equivalent to principal component. Principal component is a linear combination of the original variables that are independent (orthogonal) to other principle component. The first principal component is a new variable that accounts for the majority variation of 92.3% that corresponds to the horizontal dimension (Dim1) as shown in Fig 13. The second principal component (Dim 2) accounts for 5.5% of the total variation and is represented by the vertical axis. Together these contribute to 97.8% of the overall variation.

Fig 14 (Dim1 and Dim2) shows the cluster plot for FDEA in which the clusters are displayed in two-dimensional space. Dim 1, a new variable that accounts for 92.7% of the variation, relates to the horizontal dimension, while Dim2, which accounts for 5.2% of the variation, corresponds to the vertical axis. This accounts for 97.9% of the total variation. This

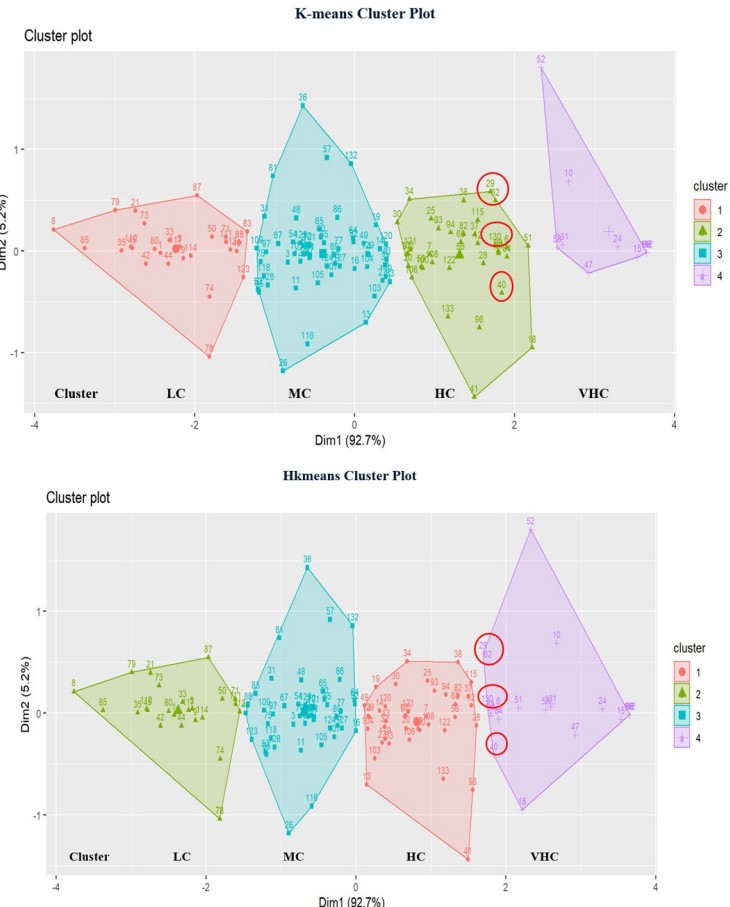

**Fig 14. Comparison of k-means and hkmeans clustering plots for FDEA.**

shows that the total variation in FDEA cluster plot has been increased by 0.1% as compared to the DEA cluster plot.

In summary, Figs 6–8 show hierarchical clusters for a specific year (2018, 2019, or 2020), whereas Fig 9 shows the difference between hkmeans clustering for DEA and FDEA datasets. Figs 11 and 12 are the comparisons between hierarchical and hkmeans clustering results for DEA and FDEA throughout the combination of all the three years. Besides that, Figs 13 and 14 show the comparisons between partitional k-means and hkmeans clustering results for both DEA and FDEA. As illustrated in Figs 6–8, the k-means algorithm has an issue in determining the k-value, whereas the hierarchical method overestimates the clustering, which does not provide a good conclusion based on the diagrams. These problems can be treated by firstly implementing the hkmeans clustering, where the hierarchical strategy is used to determine the k-value, then proceeding with the k-means strategy to generate the data clusters. Hence, this study highlights the significance of the hkmeans clustering technique to improve the drawbacks in individual partitional k-means and hierarchical algorithms so that better clustering results in terms of consistency for general data types and non-overlapping data composition for each cluster can be produced.

Table 2 displays the composition of 133 countries grouped under present four clusters of seaport network efficiency using k-means, hierarchical and hkmeans strategies imposed on DEA and FDEA datasets. It shows that the results of the k-means strategy are exactly similar

**Table 2. Composition of the countries under four seaport efficiency clusters using k-means, hierarchical and hkmeans clustering strategies.**

| Cluster | K-MEANS | | HIERARCHICAL | | HKMEANS | |
|---|---|---|---|---|---|---|
| | DEA | FDEA | DEA | FDEA | DEA | FDEA |
| LC | 41 | 41 | 50 | 55 | 21 | 24 |
| | 30.83% | 30.83% | 37.59% | 41.35% | 15.79% | 18.05% |
| MC | 52 | 14 | 12 | 7 | 49 | 46 |
| | 39.10% | 10.53% | 9.02% | 5.26% | 36.84% | 34.59% |
| HC | 14 | 52 | 70 | 70 | 40 | 40 |
| | 10.53% | 39.10% | 52.63% | 52.63% | 30.08% | 30.08% |
| VHC | 26 | 26 | 1 | 1 | 23 | 23 |
| | 19.55% | 19.55% | 0.75% | 0.75% | 17.29% | 17.29% |

between the DEA and FDEA datasets with 26 (19.55%) and 41 (30.83%) have the same value for very high connectivity (VHC) and low connectivity (LC) respectively. Some countries are clustered under medium connectivity (MC) and high connectivity (HC) with 52 (39.10%) and 14 (10.53%) for DEA whereas for FDEA it is clustered with 14 (10.53%) and 52 (39.10%) respectively which highlight the difference in the clustering. The results in this table are calculated by combination of the three years of 2018–2020 at once which is different than the yearly individual analysis done in Fig 10.

Through hierarchical clustering, it shows that 50 (37.59%) and 55 (41.35%) countries are clustered under LC for DEA and FDEA. There are significant differences in the hierarchical clustering where 56.63% and 0.75% of the countries are categorized under HC and VHC clusters for both DEA and FDEA datasets, respectively. Moreover, in comparison with the k-means and hkmeans strategies from Table 2, it is evident that the hierarchical clustering strategy produces the least composition of countries under the MC cluster with 9.02% and 5.26% for DEA and FDEA, respectively. This demonstrates that the hierarchical strategy might not be the best tool to cluster the countries associated with the seaport network efficiency due the overall imbalance composition of countries under the resulting clusters.

The percentages in Table 2 show the hkmeans clustering results with the country composition percentages of 15.79% (LC), 36.84% (MC), 30.08% (HC) and 17.29% (VHC) for DEA while 18.05% (LC), 35.34% (MC), 30.08% (HC) and 16.54% (VHC) are for FDEA. Comparing with the k-means and hierarchical clustering results, the overall country compositions under the four new seaport network efficiency clusters through the hkmeans strategy are the most balanced with minimal variation between the regular and fuzzy data distributions. This suggests the hkmeans strategy as the most recommended tool for the global seaport network efficiency clustering.

Table 3 highlights summary of the four clusters using the hkmeans strategy for FDEA dataset. Comparing with the hkmeans clustering results for DEA, the difference is minimal with only six countries namely, Brunei Darussalam, Conga, Latvia, Sierra Leone, Solomon Islands and Turkey are categorized under different cluster in DEA while the remaining 127 countries remain in the same cluster for DEA and FDEA using the hkmeans strategy. This table is selectively produced over FDEA, as a sample outcome of the hkmeans clustering method when dealing with fuzzy involvement in the dataset that may represent the real fluctuated raw data as influenced by the pandemic, economic, social, political or environmental factors.

The k-means strategy excels when dealing with clusters that are approximately spherical and have a similar number of data points. In contrast, the hierarchical clustering strategy stands out because it eliminates the need to determine the cluster count in advance, making it

**Table 3. Four levels of seaport network efficiency using FDEA of hkmeans clustering.**

| LC | MC | HC | VHC |
|---|---|---|---|
| Albania | Angola | Algeria | Argentina |
| Antigua and Barbuda | Australia | American Samoa | Bangladesh |
| Bahamas | Belgium | Barbados | Brazil |
| Bahrain | Belize | Brunei Darussalam | Cambodia |
| Cayman Islands | Benin | Bulgaria | Chile |
| Cyprus | Cameroon | Comoros | China |
| Djibouti | Canada | Croatia | China, Hong Kong SAR |
| Fiji | Colombia | Dominica | Costa Rica |
| Gabon | Congo | Egypt | El Salvador |
| Gambia | Congo (Dem. Rep. of) | Georgia | Germany |
| Guam | Côte d'Ivoire | Greece | Guatemala |
| Latvia | Cuba | Guinea-Bissau | Guinea |
| Liberia | Denmark | Guyana | India |
| Libya | Dominican Republic | Haiti | Ireland |
| Maldives | Ecuador | Iceland | Japan |
| Malta | Estonia | Iran (Islamic Rep. of) | Korea (Rep. of) |
| Mauritania | Finland | Iraq | Moldova (Rep. of) |
| Montenegro | Grenada | Italy | Netherlands |
| Mozambique | Honduras | Jamaica | Paraguay |
| Seychelles | Indonesia | Jordan | Singapore |
| Sierra Leone | Israel | Kuwait | Turkey |
| Solomon Islands | Kenya | Lebanon | United Kingdom |
| Somalia | Mexico | Lithuania | United States of America |
| Sudan | New Zealand | Madagascar | |
| | Nigeria | Malaysia | |
| | Norway | Mauritius | |
| | Pakistan | Micronesia | |
| | Peru | Morocco | |
| | Poland | Myanmar | |
| | Portugal | Namibia | |
| | Qatar | Nicaragua | |
| | Russian Federation | Oman | |
| | Saudi Arabia | Panama | |
| | Spain | Papua New Guinea | |
| | Sweden | Philippines | |
| | Tanzania | Romania | |
| | Thailand | Samoa | |
| | Timor-Leste | Senegal | |
| | United Arab Emirates | Sri Lanka | |
| | Viet Nam | Suriname | |
| | | Togo | |
| | | Tonga | |
| | | Trinidad and Tobago | |
| | | Tunisia | |
| | | Turkey | |
| | | Ukraine | |
| | | Uruguay | |

ideal for situations where the optimal number of clusters is uncertain. On the other hand, the hkmeans strategy proves its worth in handling intricate data scenarios. It is effective when addressing data with clusters at varying scales or when there's a need to explore both local and global structures within the data. In this study, four novel clusters which are low, medium, high, and very high connectivities (LC, MC, HC and VHC) have been introduced to better comprehend a country's efficiency category among the global seaport countries. The adoption of the hkmeans strategy in this specific context that was previously unexplored, has revealed new opportunities and potential outcomes for sustainable future works. Hopefully, these discoveries can empower effective decision-making and policy formulation especially the maritime industry framework.

The profound insight from the empirical results of the countries grouped under LC, is that their geographical locations are isolated or restricted from major transportation routes access because most of the countries are islands. Being situated in the remote and secluded areas exposed the countries to naturally challenging terrains and insufficient infrastructure investment opportunities in transportation networks. Consequently, such countries may encounter difficulties in establishing robust connections to global markets through seaports, airports or extensive road and rail systems, resulting in their low level of connectivity as compared to other more accessible or well-connected countries under MC, HC and VHC clusters. On the other hand, the profound insight from the empirical results of the countries grouped under VHC, is that these countries typically share common characteristics such as developing or more developed countries economically, having robust technological infrastructure, experiencing high levels of digital literacy that foster strong education systems, established global integration through trade and investment as well as openness to foreign investments, continuous urbanization efforts supported by government policies, political and economical stabilities. These factors collectively create an environment where connectivity is widely accessible and essential for various aspects of modern life, including business, education, healthcare and social interactions for the countries under the presently new VHC cluster. For instance, despite being a developing country, Bangladesh is categorized under VHC due to its economic strength as one of the world's top producer and exporter of garment industries since 1989 until the present year.

## 5. Conclusion

K-means, hierarchical and hierarchical k-means (hkmeans) clustering strategies are applied in this study to categorize 133 countries based on their seaport network efficiency scores. These scores were obtained from DEA and FDEA implementations with LSCI and GDP as the output variables. Four new level clusters have been introduced and they are sufficient to group all the global seaport countries considered. Hkmeans eliminates the sensitivity issue in the k-value selection of the k-means strategy while still producing acceptably consistent results between regular and fuzzy data distributions than the hierarchical clustering strategy. Moreover, using the hkmeans strategy that combines the partitional k-means and hierarchical algorithms, the initial partitioning of the k-means strategy can be improved to generate better clustering results in terms of general data consistency and clustered data composition.

Some limitations of the study may however be addressed here:

1. The quality of clustering results depends on the availability of sufficient data.

2. If the current free and publicly accessible maritime data becomes unavailable in the future, or if variables are altered or missing, it will lead to a reduction in the dataset size.

3. Consequently, the number of seaports considered in the study will be impacted due to the reduced availability of data.

4. The finding of this study are subject to uncertainties and fluctuations of maritime data due to COVID-19 pandemic in 2020. The hkmeans newly formed clusters may change again following arising global challenges in the future with addition of new and more data.

The present work can be extended based on the existing data by employing more varieties of machine learning methods such as naive Bayes and support vector machine, supervised or unsupervised strategies. These algorithms can also be combined with other statistical techniques such as Monte Carlo and Latin Hypercube Sampling to treat random data samples while other FDEA methods based on $\alpha$-level, fuzzy ranking and probability approaches can also be explored to provide variations in the FDEA results used in the clustering strategies. To ensure the cluster analysis remains adaptable to changing conditions, any possible vigilant data monitoring system that tracks external events and frequently updates the dataset to reflect environmental shifts can be constituted. Additionally, leveraging machine learning and AI technologies to automatically fine-tune clustering models in response to incoming data and external signals, as well as enhancing the analysis ability to accommodate dynamic changes and disruptions efficiently can be implemented.

## Author Contributions

**Conceptualization:** Noor Fadiya Mohd Noor.

**Data curation:** Dineswary Nadarajan.

**Formal analysis:** Dineswary Nadarajan, Noor Fadiya Mohd Noor.

**Investigation:** Dineswary Nadarajan, Noor Fadiya Mohd Noor.

**Methodology:** Dineswary Nadarajan, Elayaraja Aruchunan.

**Project administration:** Noor Fadiya Mohd Noor.

**Resources:** Elayaraja Aruchunan.

**Software:** Dineswary Nadarajan.

**Supervision:** Noor Fadiya Mohd Noor.

**Validation:** Dineswary Nadarajan.

**Visualization:** Dineswary Nadarajan.

**Writing – original draft:** Dineswary Nadarajan.

**Writing – review & editing:** Noor Fadiya Mohd Noor.

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
