## [Decision Letter · Decision Letter 0]

15 Mar 2024

PONE-D-23-40899New Clusterization of Global Seaport Countries Based on Their DEA and FDEA Network Efficiency ScoresPLOS ONE

Dear Dr. Mohd Noor,

Thank you for submitting your manuscript to PLOS ONE. After careful consideration, we feel that it has merit but does not fully meet PLOS ONE’s publication criteria as it currently stands. Therefore, we invite you to submit a revised version of the manuscript that addresses the points raised during the review process.

**Please provide more clarification on the discussion regarding the key insights obtained from your study and how practitioners may use this study. Also, please make the literature review more streamlined to better delineate the research contributions towards existing literature.**

We look forward to receiving your revised manuscript.

Kind regards,

Sudipta Chowdhury

Academic Editor

PLOS ONE

Journal Requirements:

Whilst you may use any professional scientific editing service of your choice, PLOS has partnered with both American Journal Experts (AJE) and Editage to provide discounted services to PLOS authors. Both organizations have experience helping authors meet PLOS guidelines and can provide language editing, translation, manuscript formatting, and figure formatting to ensure your manuscript meets our submission guidelines. To take advantage of our partnership with AJE, visit the AJE website (http://aje.com/go/plos) for a 15% discount off AJE services. To take advantage of our partnership with Editage, visit the Editage website (www.editage.com) and enter referral code PLOSEDIT for a 15% discount off Editage services. If the PLOS editorial team finds any language issues in text that either AJE or Editage has edited, the service provider will re-edit the text for free.

FRGS/1/2022/SS02/SEGI/03/1

Reviewers' comments:

Reviewer's Responses to Questions

**Comments to the Author**

1. Is the manuscript technically sound, and do the data support the conclusions?

Reviewer #1: Yes

Reviewer #2: Partly

2. Has the statistical analysis been performed appropriately and rigorously? 

Reviewer #1: Yes

Reviewer #2: No

3. Have the authors made all data underlying the findings in their manuscript fully available?

Reviewer #1: Yes

Reviewer #2: Yes

4. Is the manuscript presented in an intelligible fashion and written in standard English?

Reviewer #1: Yes

Reviewer #2: Yes

5. Review Comments to the Author

Reviewer #1: Dear authors,

The work presents k-means and hierarchical approaches, using the results of Data Envelopment Analysis and Fuzzy Data Envelopment Analysis  to aggregate 133 countries according to the efficiency scores of their port networks.

The study approach makes interesting contributions to the topic and in this regard, thank you for the effort and dedication of the authors. However, some issues must be considered to provide a better understanding. Recommendations for the manuscript are as follows:

Observation  1: Abstract

The abstract provides a contextualization of the subject, some results and part of the conclusion, but does not highlight the gap explored.

Observation 2: Number the sections

At the end of the introduction, the organization of the manuscript into numbered sections is presented, which does not match the sequence of the text. On page 14 there is a number: “4.5.2. Hierarchical versus Hkmeans for DEA and FDEA”.

Observation 3: Improve the Literature Review

The manuscript presents an interesting review, however, this section could be written in a more concise way, and some sentences could be improved to make the reading flow smoother. Connections with references can be presented in a more direct way; for example, instead of “a study” the authors could already carry out the citation as done in some parts of the text. In particular, paragraph 4 could be rewritten in a way that better relates the sentences. Furthermore, some contributions highlighted at the end of this section can be used in the introduction or conclusion.

Observation 4: Standardization figures

Some figures are different and on page 13 the x-axis title overlaps the content.

Observation 5: Conclusion

The conclusion is too lengthy; the authors can only highlight the main findings of the study and the limitations.

Reviewer #2: The design of this study is commendable, data organization and analysis are well done, but the discussion section needs to be improved and mined deeply. Some questions are as below.

1. The abstract is too long and not easy to follow. Moreover, the expression "it is the first literature that" is not appropriate and requires caution.

2. The authors should state the motives of their study and tell readers why their study is important and useful to academics and practitioners.

3. The authors should strengthen their contribution and clarify what is really their contribution and what has already been done in the literature. This is very important and in the current form of the paper their contribution seems very limited.

4. Methodology: The author needs to articulate the reasons for using FDEA and its advantages, what problems the method solves. The methods are existing methods. I suggest authors provide explanations on what and why the methodology was applied in this study. Please provide the advantages of your model.

5. For the seaport network efficiency evaluation in DEA models, the author should consider some undesirable outputs.

6. The author describes many kinds of Clustering methods, but the applicability of these kinds of Clustering are all different and cannot be used together for comparison.

7. The dissection of empirical results needs to be supplemented with more profound insights. For example, economic realities corresponding to the results should be found to corroborate them.

8. Some graphical displays cannot intuitively reveal the results, such as Figure 9, Figure 12, and so on.

9. The author should also supplement the limitations of this study and suggestions for future research in the conclusion section.

6. PLOS authors have the option to publish the peer review history of their article (what does this mean?). If published, this will include your full peer review and any attached files.

Reviewer #1: No

Reviewer #2: No

---

## [Author Response · Author response to Decision Letter 0]

30 Apr 2024

Response to Reviewers

PONE-D-23-40899 : New Clusterization of Global Seaport Countries Based on Their DEA and FDEA Network Efficiency Scores

Dear Respected Editor and Reviewers,

First of all, we would like to thank all of you for giving our manuscript the opportunity to be reviewed for possible publication in the journal of PLOS ONE. 

We have improved the language and the contents in the revised manuscript as per requests by the respected reviewers. 

Reviewer 1

Dear authors,

The work presents k-means and hierarchical approaches, using the results of Data Envelopment Analysis and Fuzzy Data Envelopment Analysis to aggregate 133 countries according to the efficiency scores of their port networks. The study approach makes interesting contributions to the topic and in this regard, thank you for the effort and dedication of the authors. However, some issues must be considered to provide a better understanding. Recommendations for the manuscript are as follows:

Observation 1: Abstract

The abstract provides a contextualization of the subject, some results and part of the conclusion, but does not highlight the gap explored.

Authors’ Reply: We have added the gap explored in this study in the abstract as suggested by the reviewer (Line 4-8).

Observation 2: Number the sections

At the end of the introduction, the organization of the manuscript into numbered sections is presented, which does not match the sequence of the text. On page 14 there is a number: “4.5.2. Hierarchical versus Hkmeans for DEA and FDEA”.

Authors’ Reply: We are sorry for the inconsistence in the section numbering. We now have add the section numbering to match the sequence on the text at the end of the introduction. 

Observation 3: Improve the Literature Review

The manuscript presents an interesting review, however, this section could be written in a more concise way, and some sentences could be improved to make the reading flow smoother. Connections with references can be presented in a more direct way; for example, instead of “a study” the authors could already carry out the citation as done in some parts of the text. In particular, paragraph 4 could be rewritten in a way that better relates the sentences. Furthermore, some contributions highlighted at the end of this section can be used in the introduction or conclusion.

Authors’ Reply: Thank you for the comment and suggestion. We have improved the connection between sentences and references to be in a more direct way by mentioning the respective author names under the Literature Review section - Paragraph 3 and Paragraph 4.

Observation 4: Standardization figures

Some figures are different and on page 13 the x-axis title overlaps the content.

Authors’ Reply: The clustering figures in this manuscript are presented as follows:

a) Figures 6 to 8 are focusing on hierarchical clustering for 2018, 2019 and 2020. 

b) Figure 9 highlight hierarchical k-means clustering for DEA and FDEA datasets

c) Figure 11 and 12 demonstrate hierarchical and hkmeans clustering side by side

More explanations on the colour notation of the dendogram branches and country number have been added in Lines 5-10, Paragraph 1 of Section 4.3 and in Lines 1-14, Paragraph 2 of Section 4.4. Moreover, the technical error where the x-axis title overlaps the content of Figure 9 has been fixed.

Observation 5: Conclusion

The conclusion is too lengthy; the authors can only highlight the main findings of the study and the limitations.

Authors’ Reply: We have revised the conclusion by highlighting the main findings, limitation and future study (Reviewer 2) as suggested while reducing the length of the conclusion part from originally 786 words to presently 397 words.

Reviewer 2:

The design of this study is commendable, data organization and analysis are well done, but the discussion section needs to be improved and mined deeply. Some questions are as below.

1. The abstract is too long and not easy to follow. Moreover, the expression "it is the first literature that" is not appropriate and requires caution.

Authors’ Reply: We have shorten the abstract from originally 250 words to 216 words presently. Moreover, we have removed the expression "it is the first literature that" from the revised abstract.

2. The authors should state the motives of their study and tell readers why their study is important and useful to academics and practitioners.

Authors’ Reply: Thank you for the comment and suggestion. We have added the motivation and the importance of the study that will be useful for academics and practitioners in Line 1-18, Paragraph 5 (last Paragraph) of Section 1.

3. The authors should strengthen their contribution and clarify what is really their contribution and what has already been done in the literature. This is very important and in the current form of the paper their contribution seems very limited.

Authors’ Reply: The contributions of the study have been strengthened and clarified in the last Paragraph of Section 2 as follows: 

1) This study introduces four new level clusters specified as low connectivity (LC), medium connectivity (MC), high connectivity (HC) and very high connectivity (VHC) in clustering the seaport network efficiency scores. The nearest study by Andrade et al. [20] categorized leading Brazilian ports based on cargo throughput efficiency, dividing them into only three groups (highly efficient, moderately efficient and inefficient).

2) This study utilizes DEA and FDEA in the countries’ seaport network efficiency measurement prior to the conduct of clustering. None of the previous studies including [21] - [31] have considered both DEA and FDEA in the clustering strategy applications at all.

3) This study clusters 133 global seaport countries which is the highest number of countries considered in similar research area. Previously Nguyen and Woo [16], employed k-means cluster analysis for top 10 Southeast Asian ports’ countries.

4) This study applies k-means and hierarchical strategies as well as recommends the third strategy, hkmeans (hierarchical k-means) for seaport network efficiency clustering. In the nearest past study, k-means strategy has been applied in social network analysis within the maritime transportation context, focusing on the top 10 Southeast Asian ports [16]. 

5) The present study implements LSCI and GDP output in both DEA and FDEA computations prior to the clustering application which was never been carried out before. Previously, Chang et al., [11] has only used LSCI to perform hierarchical clustering strategy.

4. Methodology: The author needs to articulate the reasons for using FDEA and its advantages, what problems the method solves. The methods are existing methods. I suggest authors provide explanations on what and why the methodology was applied in this study. Please provide the advantages of your model.

Authors’ Reply: We have added the reason and the advantages of FDEA as well as the problems that it can solve in the Last Paragraph of Section 3.1. 

5. For the seaport network efficiency evaluation in DEA models, the author should consider some undesirable outputs.

Authors’ Reply: Thank you for the comment and suggestion. We have highlighted the undesirable outputs as follows:

a) Line 5-9, Paragraph 4 of Section 4.2 

b) Line 26-39, Paragraph 1 of Section 4.3.

6. The author describes many kinds of Clustering methods, but the applicability of these kinds of Clustering are all different and cannot be used together for comparison. 

Authors’ Reply: We have describe the different kinds of clustering methods considered in this study together with their application in the First Paragraph of Section 4.5.2.

7. The dissection of empirical results needs to be supplemented with more profound insights. For example, economic realities corresponding to the results should be found to corroborate them.

Authors’ Reply: We have added the profound insights of empirical results as suggested by the reviewer in the entire Paragraph 10 of Section 4.5.3.

8. Some graphical displays cannot intuitively reveal the results, such as Figure 9, Figure 12, and so on.

Authors’ Reply: The clustering figures in this manuscript are presented as follows:

a) Figures 6 to 8 are focusing on hierarchical clustering for 2018, 2019 and 2020. 

b) Figure 9 highlight hierarchical k-means clustering for DEA and FDEA datasets

c) Figure 11 and 12 demonstrate hierarchical and hkmeans clustering side by side

More explanations on the colour notation of the dendogram branches and country number have been added in Lines 6-10, Paragraph 1 of Section 4.3 and in Lines 1-14, Paragraph 2 of Section 4.4. 

9. The author should also supplement the limitations of this study and suggestions for future research in the conclusion section.

Authors’ Reply: We have revised the conclusion by highlighting the main findings (Reviewer 1), limitation and future study as suggested.

---

## [Decision Letter · Decision Letter 1]

27 May 2024

New Clusterization of Global Seaport Countries Based on Their DEA and FDEA Network Efficiency Scores

PONE-D-23-40899R1

Dear Dr. Noor,

We’re pleased to inform you that your manuscript has been judged scientifically suitable for publication and will be formally accepted for publication once it meets all outstanding technical requirements.

Kind regards,

Sudipta Chowdhury

Academic Editor

PLOS ONE

Additional Editor Comments (optional):

Reviewers' comments:

Reviewer's Responses to Questions

**Comments to the Author**

1. If the authors have adequately addressed your comments raised in a previous round of review and you feel that this manuscript is now acceptable for publication, you may indicate that here to bypass the “Comments to the Author” section, enter your conflict of interest statement in the “Confidential to Editor” section, and submit your "Accept" recommendation.

Reviewer #1: All comments have been addressed

Reviewer #2: All comments have been addressed

2. Is the manuscript technically sound, and do the data support the conclusions?

Reviewer #1: Yes

Reviewer #2: Yes

3. Has the statistical analysis been performed appropriately and rigorously? 

Reviewer #1: Yes

Reviewer #2: Yes

4. Have the authors made all data underlying the findings in their manuscript fully available?

Reviewer #1: Yes

Reviewer #2: Yes

5. Is the manuscript presented in an intelligible fashion and written in standard English?

Reviewer #1: Yes

Reviewer #2: Yes

6. Review Comments to the Author

Reviewer #1: After a careful analysis, I am reporting that all the points and suggestions highlighted in the previous review have been properly considered and addressed by the authors.

The modifications to the manuscript have significantly improved its clarity, depth of analysis, and relevance to the results presented. Incorporating the feedback has substantially enhanced its scientific rigor and presentation.

Reviewer #2: The authors have made substantial revisions in response to the comments and overall the paper meets the requirements for publication. However, there is also a concern that the paper uses more cluster analyses, which seems to make it difficult to clearly draw uniform conclusions, and the authors need to place further emphasis on the presentation of this part of the results.

7. PLOS authors have the option to publish the peer review history of their article (what does this mean?). If published, this will include your full peer review and any attached files.

Reviewer #1: No

Reviewer #2: No

---

## [Editor Report · Acceptance letter]

19 Jul 2024

PONE-D-23-40899R1 

PLOS ONE

Dear Dr. Mohd Noor, 

I'm pleased to inform you that your manuscript has been deemed suitable for publication in PLOS ONE. Congratulations! Your manuscript is now being handed over to our production team.

Kind regards, 

on behalf of

Dr. Sudipta Chowdhury 

Academic Editor

PLOS ONE